# TEAM: Temporal–Spatial Consistency Guided Expert Activation for MoE Diffusion Language Model Acceleration

Linye Wei [1 2]  Zixiang Luo [3]  Pingzhi Tang [1 3]  Meng Li [1 2 4]

## Abstract

Diffusion large language models (dLLMs) have recently gained significant attention due to their inherent support for parallel decoding. Building on this paradigm, Mixture-of-Experts (MoE) dLLMs with autoregressive (AR) initialization have further demonstrated strong performance competitive with mainstream AR models. However, we identify a fundamental mismatch between MoE architectures and diffusion-based decoding. Specifically, a large number of experts are activated at each denoising step, while only a small subset of tokens is ultimately accepted, resulting in substantial inference overhead and limiting their deployment in latency-sensitive applications. In this work, we propose **TEAM**, a plug-and-play framework that accelerates MoE dLLMs by enabling more accepted tokens with fewer activated experts. TEAM is motivated by the observation that expert routing decisions exhibit strong temporal consistency across denoising levels as well as spatial consistency across token positions. Leveraging these properties, TEAM employs three complementary expert activation and decoding strategies, conservatively selecting necessary experts for decoded and masked tokens and simultaneously performing aggressive speculative exploration across multiple candidates. Experimental results demonstrate that TEAM achieves up to 2.2× speedup over vanilla MoE dLLM, with negligible performance degradation. Code is released at https://github.com/PKU-SEC-Lab/TEAM-MoE-dLLM.

## 1. Introduction

Diffusion large language models (dLLMs) (Nie et al., 2025; Ye et al., 2025; Khanna et al., 2025) address limitations of autoregressive (AR) generation by adopting bidirectional attention, which enables parallelized decoding and positions dLLMs as a compelling alternative to conventional AR models. Recent advances (Wang et al., 2025; Wu et al., 2025; Fu et al., 2025; Liu et al., 2025a) with AR initialization further strengthen this paradigm by incorporating strong autoregressive training priors while remaining compatible with KV cache, achieving both higher accuracy and better inference efficiency than AR models of comparable scale.

Thanks to the advantages in training and inference efficiency as well as their strong scalability, Mixture of Experts (MoE) (Shazeer et al., 2017; Jiang et al., 2024) architectures have become dominant in state-of-the-art AR language models (Yang et al., 2025; Liu et al., 2024; Comanici et al., 2025). Initializing dLLMs from such models, like SDAR (Cheng et al., 2025) and LLaDA 2.0 (Bie et al., 2025), further enhances the effectiveness of diffusion-based decoding, reinforcing the competitiveness of this paradigm compared to the latest AR LLMs (Ma et al., 2025b; Team et al., 2025).

Nevertheless, we observe that naively integrating MoE architectures into dLLMs can substantially degrade inference efficiency. During each diffusion iteration, all tokens within a decoding block are processed in parallel under bidirectional attention, with each token independently selecting its routed experts. However, only a small subset of tokens whose confidence exceeds a predefined threshold are ultimately unmasked. Due to the heterogeneity of expert routing decisions across tokens, a single forward pass typically activates a large fraction of the available experts, leading to significant memory access and communication overhead, while yielding only a limited number of accepted tokens.

As illustrated in Figure 1, for SDAR (Cheng et al., 2025) (8 experts per token), the observed ratio of activated distinct experts to ultimately accepted tokens is substantially higher than eight in practice. This phenomenon highlights the difficulty of applying MoE architectures to dLLMs: it effectively reverts to dense-model inefficiency. While this overhead can be amortized across concurrent requests in cloud deploy-

[1]Institute for Artificial Intelligence, Peking University, Beijing, China [2]School of Integrated Circuits, Peking University, Beijing, China [3]Yuanpei College, Peking University, Beijing, China [4]Beijing Advanced Innovation Center for Integrated Circuits, Beijing, China. Correspondence to: Meng Li <meng.li@pku.edu.cn>.

*Proceedings of the $43^{rd}$ International Conference on Machine Learning*, Seoul, South Korea. PMLR 306, 2026. Copyright 2026 by the author(s).

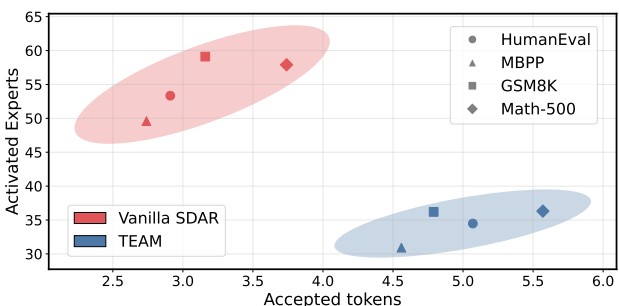

*Figure 1.* Activated experts vs. accepted tokens per forward pass in SDAR 30B-A3B. TEAM decodes more tokens with fewer experts activated in an iteration.

ments, it becomes a critical bottleneck in scenarios that are highly sensitive to decoding speed and tail latency, as well as on edge platforms with constrained hardware resources.

To address this challenge, we propose **TEAM**, which is developed based on our core observation that although both involve multi token decoding, block-wise inference in dLLMs is fundamentally different from multi batch inference in autoregressive models, exhibiting strong temporal and spatial consistency. **Along the temporal dimension**, dLLMs repeatedly process the same tokens within a block across multiple denoising levels, and for decoded tokens that have already been accepted, their values remain fixed in subsequent iterations within the block, yet they continue to participate in forward computation due to the bidirectional attention mechanism, leading to numerous unnecessary expert activations. **Along the spatial dimension**, the routing of unaccepted, masked tokens is highly concentrated, with relatively little variation in expert selection across tokens. Moreover, owing to the latent causal logic of the decoding process, the acceptance order exhibits spatial locality, suggesting that a substantial portion of masked tokens can be predicted to remain unaccepted in early iterations.

Building on these observations, TEAM implements three customized expert activation strategies for tokens within each block. For **decoded tokens**, we introduce a delayed caching mechanism, activating experts only for recently accepted tokens. For masked tokens, we further partition them into **hot tokens**, which are more likely to be accepted in the near future, and **cold tokens**, which are unlikely to be accepted. Exploiting the concentration of expert routing among masked tokens, TEAM performs speculative exploration on hot tokens to increase the acceptance rate, while rerouting cold tokens to experts that are already activated by decoded or hot tokens. Our main contributions can be summarized as follows:

- We investigate the inefficiency of naively applying

MoE architectures to dLLMs. To the best of our knowledge, this is the first study to specifically analyze expert activation characteristics in MoE dLLMs.

- Leveraging the temporal-spatial consistency of block-wise decoding in dLLMs, we propose TEAM, which implements three complementary expert activation and decoding strategies, achieving more accepted tokens with fewer activated experts.

- Extensive experiments demonstrate that, with our tailored expert activation, TEAM achieves up to 2.2× speedup while preserving model performance.

## 2. Related work

**Diffusion Large Language Models (dLLMs).** Autoregressive large language models (LLMs) (Achiam et al., 2023; Grattafiori et al., 2024; Yang et al., 2025) have demonstrated remarkable capabilities across a wide range of applications across a wide range of applications such as text generation, medical science (Zhu et al., 2025a), and embodied AI (Yang et al., 2026b), yet their inference efficiency is constrained by autoregressive decoding. Inspired by the success of diffusion in generative modeling (Ho et al., 2020; Yan et al., 2025; 2026; Zhu et al., 2025b), diffusion large language models (dLLMs) (Nie et al., 2025; Ye et al., 2025; Li et al., 2025a; Yang et al., 2026a; Feng et al., 2026) mitigate this limitation by enabling parallel decoding via bidirectional attention mechanism. In this paradigm, the entire response is represented as masked tokens, and all positions are decoded in each forward pass. Tokens whose confidence exceeds a predefined threshold are accepted, while the remaining tokens are re-masked and refined in subsequent iterations. However, the global bidirectional attention prevented reuse of KV cache, resulting in limited efficiency gains. More recent work (Wang et al., 2025; Tian et al., 2025; Arriola et al., 2025; Gong et al., 2025) adopts a block diffusion strategy, partitioning the response into multiple blocks. By retraining from autoregressive initialization, these models learn to apply bidirectional attention within a block to enable parallel decoding, while using causal attention across blocks to support KV cache. In this way, dLLMs inherit strong autoregressive priors for accuracy while simultaneously improving inference efficiency through parallel decoding and cache reuse.

**Mixture-of-Experts (MoE).** Unlike dense models which activate all parameters during each forward pass, Mixture-of-Experts (MoE) architectures (Shazeer et al., 2017; Jiang et al., 2024; Xu et al., 2026) replace a single feedforward layer with a group of parallel expert networks and employ a gating mechanism to dynamically select a subset of experts for each token. By enabling expert specialization and sparse parameter activation, MoE architectures scale more effec-

tively to achieve higher model capacity while improving both training and inference efficiency (Huang et al., 2025; Zhong et al., 2025), and have consequently become a dominant design in recent LLMs (Yang et al., 2025; Liu et al., 2024; Comanici et al., 2025). This paradigm has also been extended to dLLMs (Zhu et al., 2025c; Cheng et al., 2025; Bie et al., 2025). However, in dLLMs, all tokens within a block are processed at every diffusion iteration, and each token independently activates its routed experts. As a result, even when the batch size is one, a large fraction of the model's parameters may be activated in a single forward pass. Such scenarios are often precisely those that are most sensitive to decoding latency, thereby limiting the practical deployment of MoE-based dLLMs.

**Acceleration of dLLMs.** Several studies have explored techniques to accelerate dLLMs. Early approaches (Wu et al., 2025; Liu et al., 2025b; Ma et al., 2025a) reduce computation through approximate KV caching, but rely on fixed and coarse-grained intervals for cache refresh. Other methods (Chen et al., 2025; Song et al., 2025; Jiang et al., 2025; Qian et al., 2026) limit arithmetic intensity via sparsification, which becomes less critical for block diffusion. A separate line of work (Gao et al., 2025; Agrawal et al., 2025; Wei et al., 2025; Wu & Zhang, 2025; Zhu et al., 2025d) integrates latent refinement or speculative decoding (Chen et al., 2023; Leviathan et al., 2023) into dLLMs to further increase decoding parallelism. Beyond dense models, the strong empirical performance of MoE dLLMs has recently motivated efforts to accelerate this paradigm, such as dInfer (Ma et al., 2025b). However, dInfer primarily targets general dLLM acceleration and focuses only on expert-parallel execution for cloud-scale MoE deployment. In contrast, we present the first dedicated analysis of expert activation behavior in MoE dLLMs and propose TEAM, which exploits temporal–spatial consistency to tailor distinct expert activation strategies for different tokens, thereby improving decoding efficiency.

## 3. Preliminary and Motivation

The inference process of a dLLM begins by initializing the response $Y$ with $N = B \times L$ [MASK] tokens, where the response is partitioned into $B$ blocks, each containing $L$ tokens. The $i$-th block is denoted as $Y_i = \left[y_i^0, y_i^1, \cdots, y_i^{L-1}\right]$. Given a prompt $P$, the model $p_\vartheta$ performs block-wise decoding, factorizing the response $\widehat{Y}$ as:

$$p_\vartheta(\widehat{Y} \mid P) = \prod_{i=1}^{B} p_\vartheta\left(\widehat{Y}_i \mid P, Y_{\leq i}\right) \qquad (1)$$

Concretely, within each block, the model iteratively samples from the [MASK] tokens. For the $i$-th block, a single forward pass produces token predictions and corresponding confidence scores, given by:

$$
\begin{aligned}
\widehat{y_i^k} &= \operatorname*{argmax}_{v \in V} p_\vartheta\left(y_i^k = v \mid P, Y_{\leq i}\right) \text{ and} \\
c_k &= p_\vartheta\left(y_i^k = \widehat{y_i^k} \mid P, Y_{\leq i}\right), k \in [0, 1, \cdots, L]
\end{aligned}
\qquad (2)
$$

where $V$ denotes the model's vocabulary. Only tokens whose confidence exceeds a predefined threshold $\tau$ are accepted, while the remaining tokens are re-masked for subsequent iterations:

$$y_i^k = \begin{cases} \widehat{y_i^k}, & \text{if } c_k > \tau \\ [\text{MASK}], & \text{otherwise} \end{cases}, k \in [0, 1, \cdots, L] \quad (3)$$

Once all positions within a block are unmasked, the block is cached and decoding proceeds to the next, repeating this process until the end-of-sequence [EOS] token is generated.

The decoding efficiency of dLLMs arises from the parallel processing of all tokens within a block, which allows multiple tokens to be accepted in a single iteration. However, when this block-wise decoding paradigm is combined with MoE architectures, the parallel tokens collectively activate a large fraction of the experts, negating the benefits of sparse parameter activation. As a result, the combined system can underperform relative to the theoretical gains of its constituent components. To better understand this inefficiency, we analyze the decoding trajectory of a single block along with its associated expert activation patterns, as illustrated in Figure 2, from which we derive several key observations.

**Temporal Consistency.** Block-wise decoding in diffusion large language models (dLLMs) requires repeatedly processing the same block across successive denoising iterations, during which tokens are gradually accepted and propagated to subsequent steps. Although these accepted tokens no longer change and merely provide context for decoding the remaining masked tokens, they still incur full computation at every iteration and independently trigger expert activations in MoE layers. Figure 2(a) reports the number of experts activated across iterations at three representative layers (first, middle, and last). This repeated computation on already decoded tokens leads to substantial additional expert activations at each step. As decoding progresses and more tokens become fixed, the fraction of total activated experts attributable to these tokens increases, ultimately dominating expert activation in later iterations.

**Spatial Consistency.** In contrast to the token-specific activations of decoded tokens, variability among masked tokens primarily stems from positional encodings in their input embeddings. Consequently, spatially adjacent masked tokens

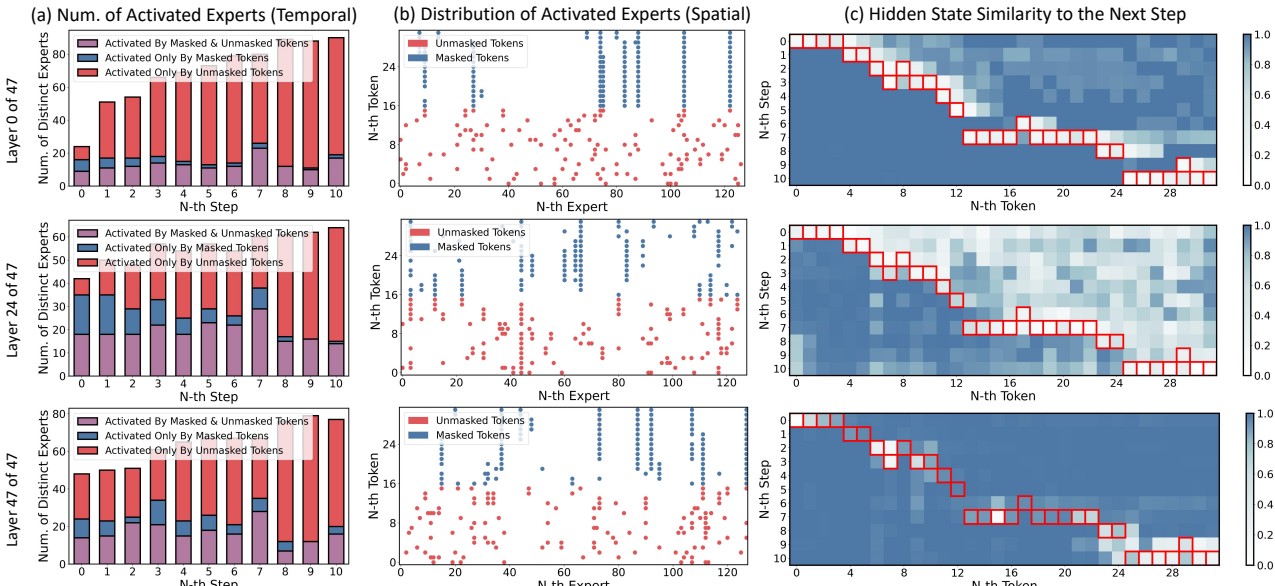

*Figure 2.* Temporal-spatial characteristics of expert activation and decoding with the SDAR 30B-A3B model on a prompt from the GSM8K dataset. Results are shown for layers 0, 24, and 47 (of 47). (a) Number of activated experts across decoding iterations. (b) Distribution of experts activated by decoded and masked tokens at step 6 (of 11). (c) Token acceptance positions at each iteration, together with hidden state similarity relative to the subsequent iteration. The generality across more datasets is shown in Appendix A.1.

exhibit highly consistent expert routing patterns across layers. As shown in Figure 2(b), while decoded tokens activate a diverse set of experts with an approximately uniform distribution across candidates, masked tokens tend to concentrate their routing decisions on a small subset of experts. This observation suggests that a specific group of experts dominates the decoding of nearly all masked tokens, whereas experts outside this subset contribute only marginally or are invoked by very few tokens. The spatial concentration of expert activation also explains why, in Figure 2(a), the experts activated by masked tokens constitute only a limited fraction of the total activated experts.

**Temporal-Spatial Locality.** We further analyze the step-wise similarity of hidden states produced by each layer and mark the token positions that are unmasked at each iteration (highlighted by red boxes), as illustrated in Figure 2(c). We find that the most significant change in the hidden state of a token occurs precisely between the iteration in which it is accepted and the subsequent iteration, consistent with observations in prior work (Ma et al., 2025a; Song et al., 2025; Li et al., 2025b). Once a token has been accepted and processed through one additional forward pass, its representation can be regarded as approximately stable. Moreover, the acceptance order within a block largely follows an autoregressive trend over time, and the tokens accepted at each iteration exhibit spatial clustering. This behavior is expected, given that dLLMs are initialized from autoregressive models and that natural language generation inherently follows a causal

structure. As a result, tokens newly accepted at a given step tend to lie close to previously decoded tokens, while some tokens are unlikely to be accepted during early iterations.

Building on these insights, we propose TEAM, which implements three complementary expert activation and decoding strategies tailored to the tokens within each block, as shown in Figure 3.

## 4. TEAM Methodology

### 4.1. Delayed Caching for Decoded Tokens (DCD)

As discussed before, once decoded tokens are incorporated as input and processed by an additional forward pass after being accepted, their hidden representations become approximately stable. This observation naturally motivates caching these tokens to avoid redundant computation across iterations. Specifically, at each iteration, computation is performed only for the masked tokens and the tokens newly accepted in the previous step, while the key–value pairs of tokens decoded in earlier iterations are reused from cache. After each forward pass, the KV pairs corresponding to newly accepted tokens are inserted into the cache and reused in subsequent iterations.

A related strategy has been explored in prior work dKV-Cache (Ma et al., 2025a) targeting global bidirectional attention. To mitigate KV drift of decoded tokens under bidirec-

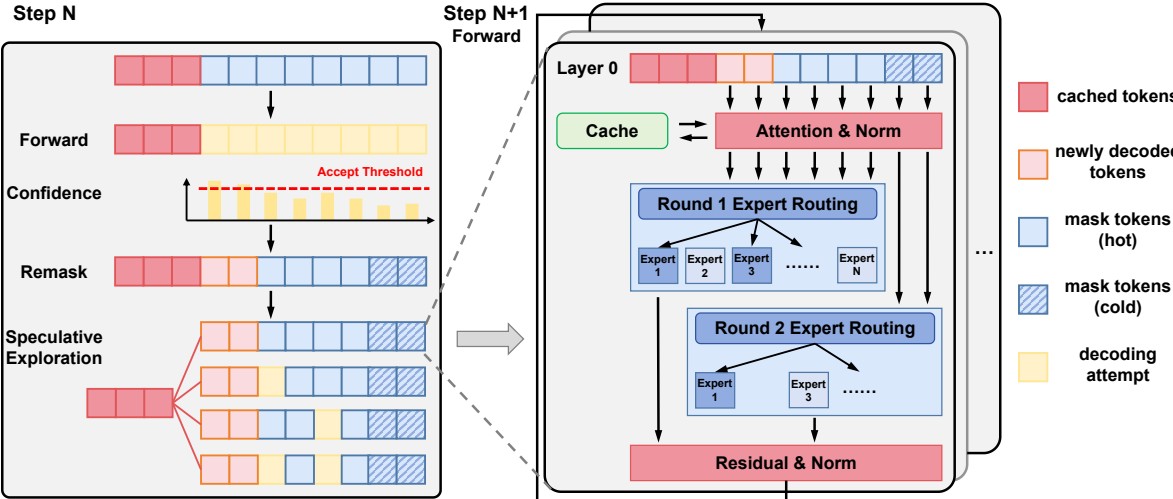

*Figure 3.* Overview of our proposed **TEAM**. We apply differentiated expert activation and decoding strategies to tokens within each block. For **decoded tokens**, redundant computation is reduced through one-step delayed caching. For **mask tokens (hot)**, we adopt aggressive multi-branch speculative exploration to exploit idle compute resources and increase the token acceptance rate. For **mask tokens (cold)**, a double-round routing mechanism is introduced to constrain unnecessary expert activations.

tional attention, dKV-Cache additionally performs periodic global cache refresh by recomputing all tokens every $N$ iterations. This mechanism becomes less effective in dLLMs that adopt the block diffusion paradigm with native support for KV cache. In this setting, decoding no longer requires global token processing, and parallel computation is confined to tokens within a single block. Such block-level parallelism typically does not reach the compute–bandwidth balance point of modern hardware platforms such as GPUs, leaving decoding largely memory-bound and rendering fine-grained KV caching within a block unnecessary.

However, as observed earlier, under MoE architectures, decoded tokens activate a large set of experts that are largely distinct from those activated by masked tokens, substantially increasing parameter activation density and memory access. This property makes caching decoded tokens particularly beneficial in MoE-based dLLMs. Moreover, by leveraging autoregressive priors and the near-autoregressive acceptance order during decoding, our delayed caching mechanism eliminates the need for periodic global cache refresh. As demonstrated in Section 5, this design improves decoding efficiency without sacrificing model quality.

### 4.2. Speculative Exploration for Hot Tokens (SEH)

Beyond eliminating redundant expert activations arising from repeatedly processing decoded tokens, we observe that expert activation and decoding for masked tokens are also inefficient. First, each activated expert is typically re-

sponsible for only a small number of tokens, resulting in frequent memory accesses with low computational intensity. Consequently, available compute resources are underutilized, increasing overall decoding latency. Second, the near-autoregressive decoding order and the spatial clustering of accepted tokens indicate that a subset of tokens has a low probability of being accepted in early iterations. Activating experts for such tokens only to remask them afterward leads to wasted computation.

These observations motivate a differentiated computation strategy for masked tokens: we aim to improve decoding efficiency for tokens that are likely to be accepted in the near future, which we refer to as **hot tokens**, while reducing the computational overhead for tokens that are unlikely to be accepted, referred to as **cold tokens**. We identify two characteristics that make masked tokens more likely to be accepted in the next iteration: (1) their decoding attempt $y_i^k$ at the current iteration yields a relatively high confidence score $c_k$, even if it does not yet exceed the acceptance threshold; and (2) they are spatially closer to decoded tokens, and thus benefit from more informative contextual guidance. Formally, the hot tokens are defined as:

$$y_i^{k-hot} = \left\{ y_i^k \mid (c_k > \tau_h) \text{ or } (\forall j, |k - j| < L_h) \right\} \quad (4)$$

where $\tau_h$ denotes the confidence threshold to identify hot tokens, $j$ indexes the positions of currently decoded tokens, and $L_h$ specifies the maximum allowable distance from

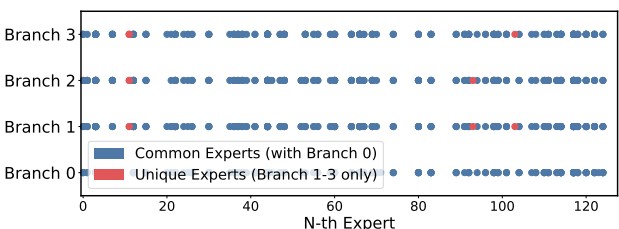

*Figure 4.* Expert activation with speculative exploration in SDAR for a response from the GSM8K dataset, measured at layer 24 (of 47). More cases are provided in Appendix A.2.

decoded tokens for a masked token to be classified as hot.

For these hot tokens, we employ speculative exploration by additionally accepting the tokens with the top-$k$ confidence scores to construct multiple branches. By decoding and verifying these branches in parallel, we increase the acceptance rate at each iteration, thereby reducing the total number of decoding steps and overall latency. Under bidirectional attention, modifying any single token affects all tokens within the block, implying that each additional candidate effectively incurs the full computational cost of an entire block. For dense models, this behavior is prohibitive, as it rapidly increases computational intensity and shifts the decoding bottleneck from memory-bound to compute-bound, often requiring multi-GPU parallelism to achieve meaningful speedups (Xu et al., 2025). In contrast, for MoE architectures, inference latency is dominated by feedforward layers, and computation is naturally distributed across experts. As illustrated in Figure 4, introducing additional decoding branches increases the originally low arithmetic intensity of each expert, while the high similarity across branches avoids activating a large number of new experts. Consequently, our exploration remains effective for MoE architectures even on a single GPU.

### 4.3. Limited Activation for Cold Tokens (LAC)

Cold tokens are defined as masked tokens that are both distant from decoded token positions and associated with low confidence scores in previous decoding attempts. Such tokens are unlikely to be accepted in subsequent iterations, implying that activating experts uniquely routed to them is often unnecessary. In practice, decoding attempts for cold tokens have a high probability of being re-masked after each forward pass, rendering their dedicated expert activations largely wasteful.

Leveraging the spatial consistency, masked tokens tend to route to a largely shared subset of experts, with only minor variation across tokens. Based on this property, we apply a limited activation strategy for cold tokens, as summarized

---

**Algorithm 1** Limited Activation (Appendix B)

**Input:** Decoded tokens $D$, Mask tokens $M$, Experts $E_0$
**Output:** Activated Experts $E_A$, Routing Weights $W$
**// 1. Classfication of tokens**
Find newly accepted tokens $D_a \subseteq D$
Find hot tokens $H \subseteq M$ via **Eq. 4**
Define cold tokens $C \leftarrow M \setminus H$
**// 2. First-round Routing**
Necessary activation $W_1 \leftarrow Router(D_a, H, E_0)$
Necessary experts $E_A \leftarrow$ top-$k$ $(W_1)$
**// 3. Second-round Routing**
Activation for cold tokens $W_2 \leftarrow Router(C, E_A)$
Routing Weights $W \leftarrow Concat(W_1, W_2)$
**Return** $E_A, W = 0$

---

in Algorithm 1. Specifically, we first perform expert routing for newly accepted tokens that have not yet been cached, as well as for hot tokens, since these tokens require accurate expert activation to ensure decoding correctness. The union of experts selected in this stage defines a necessary expert set. We then conduct a second round of routing for cold tokens, restricting their activation to this expert set. Through this double-round routing mechanism, expert activation for cold tokens is strictly confined to necessary experts, avoiding the introduction of token-specific expert activations while preserving model quality and retaining the possibility that cold tokens may still be unexpectedly accepted in later iterations.

## 5. Experiments

### 5.1. Experimental Setup

Our experiments are primarily conducted on SDAR 30B-A3B (Cheng et al., 2025), a representative MoE-based diffusion language model that follows the block diffusion paradigm. We adopt the official open-sourced evaluation protocol provided by SDAR as our baseline. We note that for another MoE-based dLLM, LLaDA 2.0 (Bie et al., 2025), an official Hugging Face–format evaluation pipeline is not available, and therefore it is not used as the primary experimental platform in this study. To demonstrate the generality of our proposed TEAM across diverse tasks, we evaluate performance on two code generation benchmarks, HumanEval (Chen et al., 2021) and MBPP (Austin et al., 2021), as well as two mathematical reasoning benchmarks, GSM8K (Cobbe et al., 2021) and Math-500 (Lightman et al., 2023). All experiments are conducted on an NVIDIA A100 80GB GPU.

Unless otherwise specified, we follow the official implementations of SDAR and LLaDA 2.0. The acceptance threshold for unmasking tokens is set to $\tau = 0.95$, and the block size, which is the number of tokens decoded in parallel at each iteration, is fixed to 32. For masked token classification

*Table 1.* **Performance of TEAM on SDAR.** APF denotes the number of Activated experts Per Forward pass, TPF denotes accepted Tokens Per Forward pass, and APT denotes the equivalent number of Activated experts Per decoded Token.

| Benchmark | Method | Score↑ | APF↓ | TPF↑ | APT↓ | Speedup |
|---|---|---|---|---|---|---|
| HumanEval | Vanilla | 79.27 | 53.34 | 2.91 | 18.33 | 1× |
| *(0-shot)* | TEAM | 79.88 (+0.61) | 34.48 (35%↓) | 5.07 (1.74×) | 6.80 (63%↓) | 2.20× |
| MBPP | Vanilla | 65.76 | 49.59 | 2.74 | 18.10 | 1× |
| *(0-shot)* | TEAM | 65.76 (+0.00) | 30.92 (38%↓) | 4.56 (1.66×) | 6.78 (63%↓) | 2.08× |
| GSM8K | Vanilla | 90.60 | 59.11 | 3.16 | 18.71 | 1× |
| *(0-shot)* | TEAM | 90.30 (−0.30) | 36.20 (39%↓) | 4.79 (1.52×) | 7.56 (60%↓) | 1.83× |
| Math-500 | Vanilla | 76.00 | 57.90 | 3.74 | 15.48 | 1× |
| *(0-shot)* | TEAM | 75.40 (−0.60) | 36.31 (37%↓) | 5.57 (1.49×) | 6.52 (58%↓) | 1.64× |
| Average | Vanilla | 77.91 | 54.99 | 3.14 | 17.66 | 1× |
|  | TEAM | 77.84 (−0.07) | 34.48 (37%↓) | 5.00 (1.59×) | 6.92 (61%↓) | 1.94× |

*Table 2.* **Sensitivity Analysis of Hot-Token Hyperparameters.** Model accuracy and the number of Activated experts Per Forward pass (APF) with DCD and LAC strategy on code generation benchmarks.

| Benchmark | $\tau_h = 0.4, L_h = 6$ | | $\tau_h = 0.5, L_h = 5$ | | $\tau_h = 0.6, L_h = 4$ | | $\tau_h = 0.7, L_h = 3$ | | $\tau_h = 0.8, L_h = 2$ | |
|---|---|---|---|---|---|---|---|---|---|---|
| | Score | APF | Score | APF | Score | APF | Score | APF | Score | APF |
| HumanEval | 78.05 | 24.49 | 81.09 | 24.38 | 79.27 | 24.01 | 79.27 | 23.08 | 77.44 | 22.26 |
| MBPP | 66.93 | 22.21 | 65.76 | 22.15 | 68.09 | 22.12 | 66.93 | 21.66 | 62.65 | 20.40 |
| Average | 72.49 | 23.35 | 73.43 | 23.27 | 73.68 | 23.07 | 73.10 | 22.37 | 70.05 | 21.33 |

in TEAM, we identify hot tokens either by a confidence threshold $\tau_h = 0.7$ or by requiring them to be no more than $L_h = 3$ positions away from decoded tokens. In speculative exploration for hot tokens, the number of parallel branches is set to 4.

## 5.2. Main Results

We report the improvements of TEAM over the vanilla model in terms of expert activation and decoding efficiency, as summarized in Table 1. Without TEAM, each layer of the vanilla model activates more than 50 experts on average per forward pass, approaching half of the total 128 experts. However, each forward pass accepts and unmasks only approximately three tokens, leading to more than twice the nominal routing cost (8 experts per token) for decoding a single token, and in the worst case up to 18 activated experts per decoded token. This reveals a fundamental inefficiency: although parallel decoding in dLLMs and sparse parameter activation in MoE architectures are both individually designed to be inference efficient, their naive combination becomes counterproductive. This inherent incompatibility significantly degrades the overall decoding speed.

In contrast, TEAM achieves decoding of more tokens with substantially fewer expert activations through its carefully designed expert activation and decoding strategies, thereby

simultaneously realizing the benefits of parameter sparsity and decoding parallelism. By introducing delayed caching for decoded tokens and limited expert activation for cold tokens, TEAM reduces the number of activated experts by 35–39% across all benchmarks, even after accounting for the additional experts introduced by multi-branch exploration. Moreover, speculative exploration for hot tokens further enhances decoding parallelism, increasing the number of accepted tokens per iteration by 1.49–1.74×. Benefiting from both higher sparsity and increased parallelism, TEAM requires on average only 6.92 activated experts to decode a single token, which is even lower than the nominal routing cost of 8 experts per token. As a result, TEAM achieves an average speedup of 1.94×, with a peak speedup of up to 2.2× on the HumanEval benchmark.

## 5.3. Ablation Study and Hyperparameter Sensitivity

To assess the contribution of TEAM to accelerating inference in MoE dLLM, we progressively integrate its core techniques into the vanilla model. As illustrated in Figure 5, we evaluate both the average number of activated experts required to decode a single token and the corresponding speedup throughout this process.

The ablation study show that the introduction of Speculative Exploration for Hot Tokens (SEH) substantially reduces the

*Table 3.* **Performance with Delayed Caching for Decoded Tokens (DCD).** Refresh-4 denots block cache refresh every 4 steps, Refresh-8 denots block cache refresh every 8 steps, and Refresh-free denotes never refresh (Ours).

| Benchmark | Refresh-4 | | | Refresh-8 | | | Refresh-free | | |
|---|---|---|---|---|---|---|---|---|---|
| | Score | APF | Speedup | Score | APF | Speedup | Score | APF | Speedup |
| HumanEval | **79.88** | 32.67 | 1.38× | 78.66 | 29.61 | 1.47× | **79.88** | 26.72 | 1.58× |
| MBPP | **66.15** | 29.89 | 1.32× | **66.15** | 26.91 | 1.44× | 65.76 | 23.67 | 1.55× |
| GSM8K | 90.52 | 35.45 | 1.27× | **90.60** | 31.23 | 1.35× | 90.45 | 27.99 | 1.44× |
| Math-500 | **74.80** | 35.25 | 1.17× | 73.00 | 31.18 | 1.25× | 74.20 | 27.52 | 1.32× |
| Average | 77.84 | 33.32 | 1.29× | 77.10 | 29.73 | 1.38× | 77.57 | 26.48 | 1.47× |

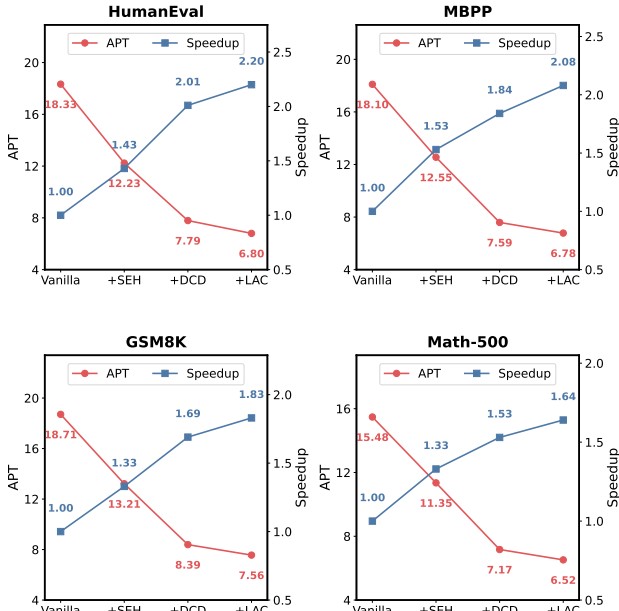

*Figure 5.* Ablation study on the Activated experts Per decoded Token (APT) and speedup compared to the vanilla model.

number of activated experts per decoded token, validating its effectiveness. By decreasing the number of decoding steps, SEH increases the token acceptance rate per forward pass, while incuring only a marginal increase in additional expert activations thanks to the similarity across multiple branches. Delayed Caching for Decoded Tokens (DCD) further eliminates a large fraction of expert activations triggered by already decoded tokens that are irrelevant to the decoding process itself and only provide contextual guidance. Finally, Limited Activation for Cold Tokens (LAC) strictly confines expert activation to the subset responsible for newly decoded tokens and hot tokens. This design further reduces the number of activated experts per decoded token and yields additional speedup, resulting in the highest overall decoding efficiency among all benchmarks.

Furthermore, we will discuss the hyperparameter selection in our proposed TEAM.

A key factor in accelerating MoE dLLMs with TEAM is the criterion used to classify masked tokens into hot and cold categories, which is governed by a confidence threshold $\tau_h$ and a distance constraint $L_h$. We study the impact of different combinations of these two hyperparameters on model accuracy and the number of activated experts per forward pass on two code generation benchmarks, HumanEval and MBPP, as summarized in Table 2. As $\tau_h$ increases up to 0.7 and $L_h$ decreases to 3, the model preserves its accuracy while the number of activated experts decreases monotonically, indicating improved efficiency. Further narrowing the set of tokens identified as hot leads to noticeable performance degradation. Consequently, the configuration $\tau_h = 0.7$ and $L_h = 3$ achieves the best accuracy-efficiency trade-off and is adopted as the default setting in TEAM.

### 5.4. Analysis of Key Design Choices in TEAM

**Refresh-free Caching for Decoded Tokens.** As discussed in Section 4, caching decoded tokens at every iteration like dKV-Cache yields limited acceleration for dLLMs that adopt the block diffusion paradigm. However, reducing expert activations through caching is critical to the efficiency of MoE dLLMs. Unlike prior work that periodically refreshes cached representations, we observe that for dLLMs initialized from autoregression, the representations of previously decoded tokens remain highly stable across diffusion iterations, rendering cache refresh unnecessary. As shown in Table 3, we evaluate different refresh intervals, including recomputing all tokens every 4 steps, every 8 steps, or disabling refresh entirely within a block. We find no evidence that eliminating cache refresh leads to a noticeable degradation in model performance. In fact, the Refresh-free configuration even achieves slightly higher accuracy than Refresh-8. In contrast, the efficiency gains from removing refresh operations are substantial, making refresh-free caching a clear advantage for MoE dLLM inference.

**Aligned Token Candidates in SEH.** To fully exploit the

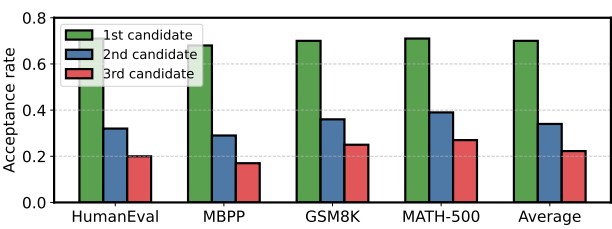

*Figure 6.* Acceptance probability of the top3-confidence candidate tokens in the subsequent iteration in SEH.

*Table 4.* Highest and average pairwise cosine similarity of hidden states among all masked tokens within a block across different layers.

|  | **HumanEval** | **MBPP** | **GSM8K** | **MATH-500** |
|---|---|---|---|---|
| Highest | 0.99 | 0.98 | 0.99 | 0.99 |
| Average | 0.86 | 0.84 | 0.86 | 0.86 |

idle compute resources on GPU platforms, we introduce speculative exploration through additional acceptance of candidate tokens. Specifically, three extra branches are constructed by additionally accepting the 1st-confidence token, the 2nd-confidence token, and both tokens simultaneously, even when their confidence scores do not exceed the acceptance threshold. We argue that this design is preferable to further accepting the 3rd-confidence token. To justify this choice, we analyze the acceptance probability of the top3-confidence candidate tokens in the subsequent iteration, as illustrated in Figure 6. The 2nd candidate has a relatively low probability of being directly accepted, while the 3rd one is rarely accepted. This indicates that constructing more diverse branches is inefficient. In contrast, aligned token combinations enable chained verification, thereby improving efficiency.

**The Shared Expert Subset for Masked Tokens.** A key observation of our proposed TEAM is that a specific subset of experts dominates the decoding process for most masked tokens, while the remaining experts contribute only marginally or are rarely activated. We further investigate the underlying reason for this phenomenon and provide evidence to support it. The dominance arises from the high similarity among model inputs for masked tokens. All masked tokens share the same [M] symbol and are mapped to identical token embeddings. The only source of variation comes from positional encodings, which are relatively minor within a block.As shown in Table 4, we compute the average pairwise cosine similarity of hidden states among all masked tokens within a block at each decoding layer, and then report the highest and the average similarity in all layers. Such a high degree of similarity indicates that this homogeneity persists throughout the entire decoding process, leading to similar routing decisions.

## 6. Conclusion

In this paper, we propose **TEAM**, a framework for accelerating MoE diffusion language models through temporal and spatial consistency guided expert activation. Through a systematic analysis of temporal consistency across denoising iterations and spatial consistency across token positions, TEAM introduces three complementary expert activation and decoding strategies tailored to different subsets of tokens within each decoding block, enabling more tokens to be decoded with fewer activated experts. Compared to vanilla inference, TEAM achieves up to a 2.2× speedup, demonstrating an efficient and practical integration of dLLM and MoE architectures.

## Acknowledgments

This work was supported in part by the National Key Research and Development Program under Grant 2024YFB4505004, in part by NSFC under Grant 62495102, Grant 92464104, and Grant 62341407, in part by Beijing Municipal Science and Technology Program under Grant Z241100004224015, in part by 111 Project under Grant B18001.

## Impact Statement

This paper presents work whose goal is to advance the field of Machine Learning. There are many potential societal consequences of our work, none which we feel must be specifically highlighted here.

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

# A. More details of Temporal-spatial characteristics in dLLMs

## A.1. Temporal-spatial characteristics of expert activation and decoding

To demonstrate the generality of our observations, this section presents the temporal–spatial characteristics of expert activation and decoding in the SDAR 30B-A3B model across prompts from diverse benchmarks. We find that MoE-based diffusion language models consistently exhibit strong temporal consistency across denoising iterations and spatial consistency across token positions, which directly motivates the design of our proposed TEAM.

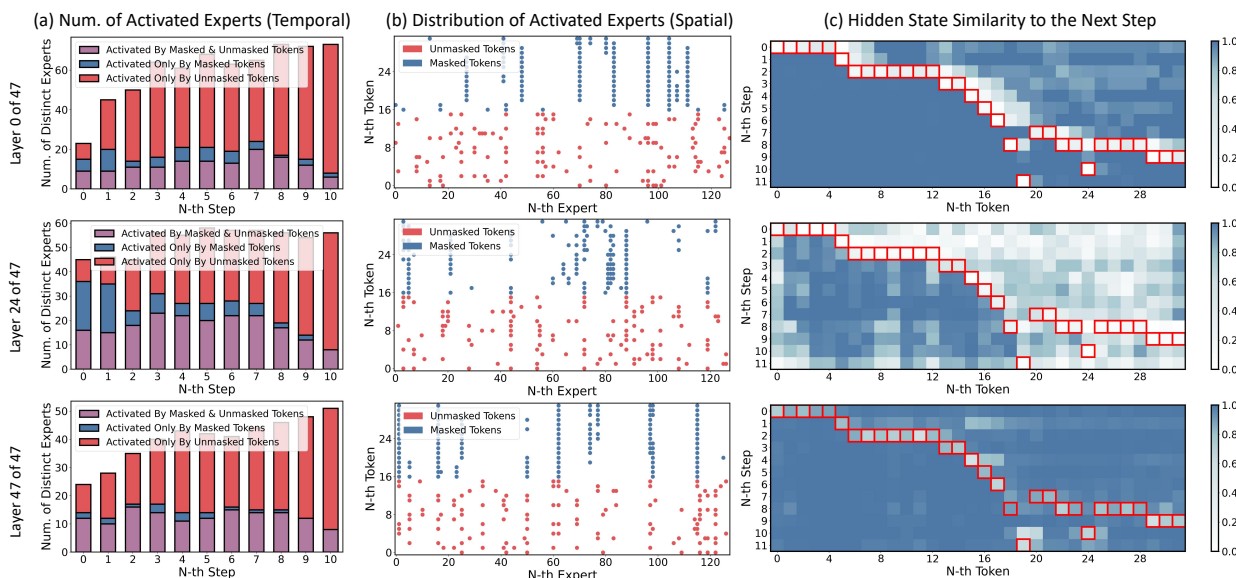

*Figure 7.* Temporal-spatial characteristics of expert activation and decoding with the SDAR 30B-A3B model on a prompt from the HumanEval dataset.

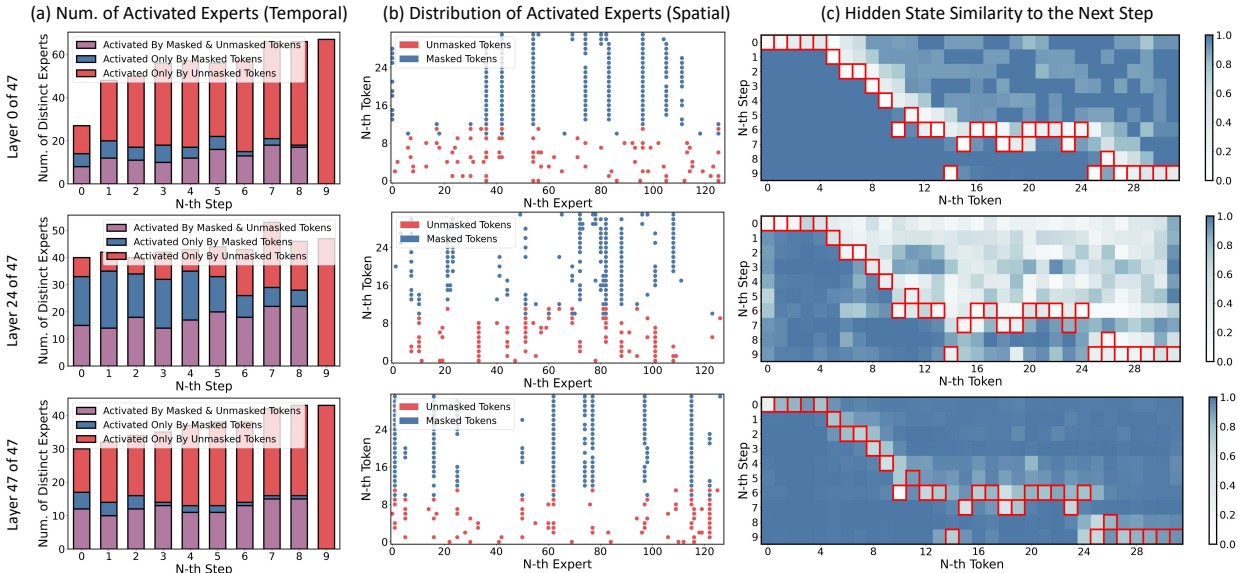

*Figure 8.* Temporal-spatial characteristics of expert activation and decoding with the SDAR 30B-A3B model on a prompt from the MBPP dataset.

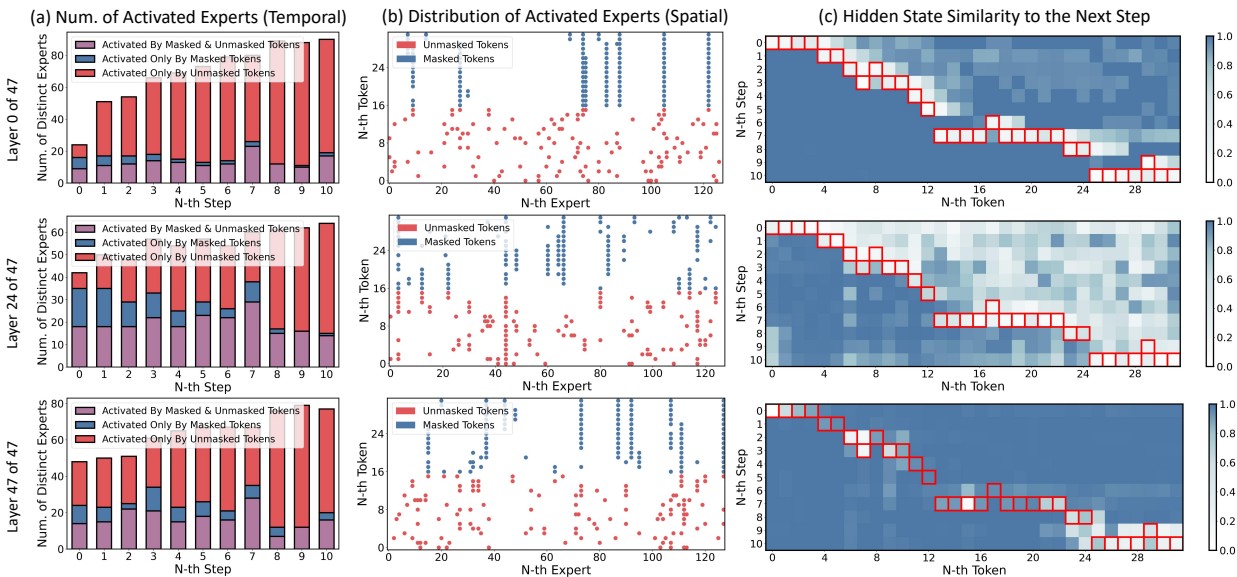

*Figure 9.* Temporal-spatial characteristics of expert activation and decoding with the SDAR 30B-A3B model on a prompt from the GSM8K dataset.

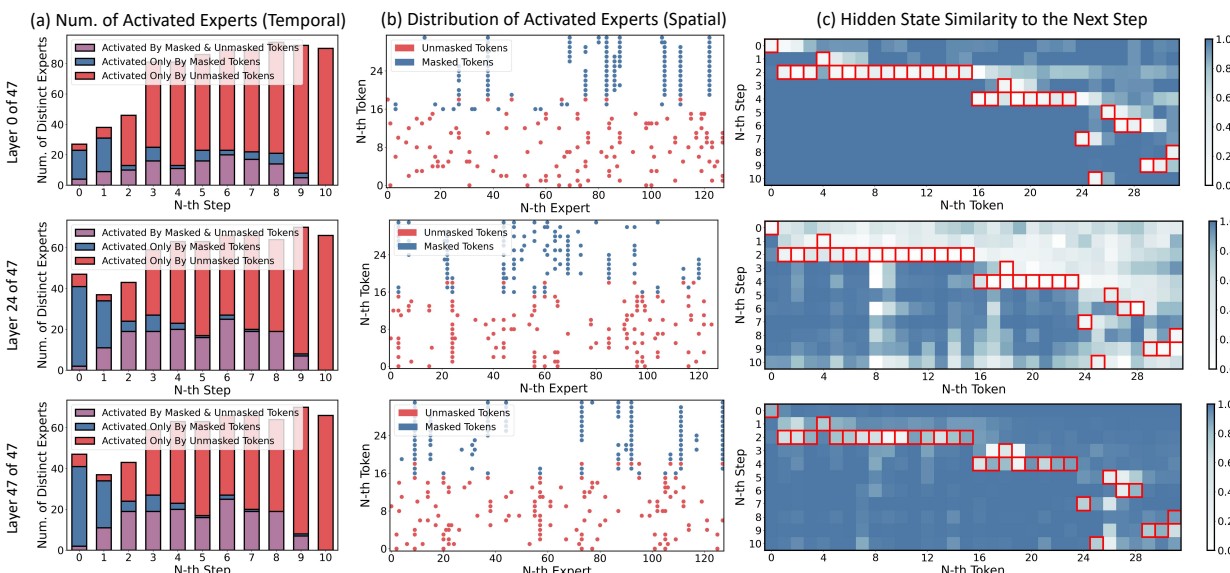

*Figure 10.* Temporal-spatial characteristics of expert activation and decoding with the SDAR 30B-A3B model on a prompt from the Math-500 dataset.

## A.2. Expert activation with speculative exploration

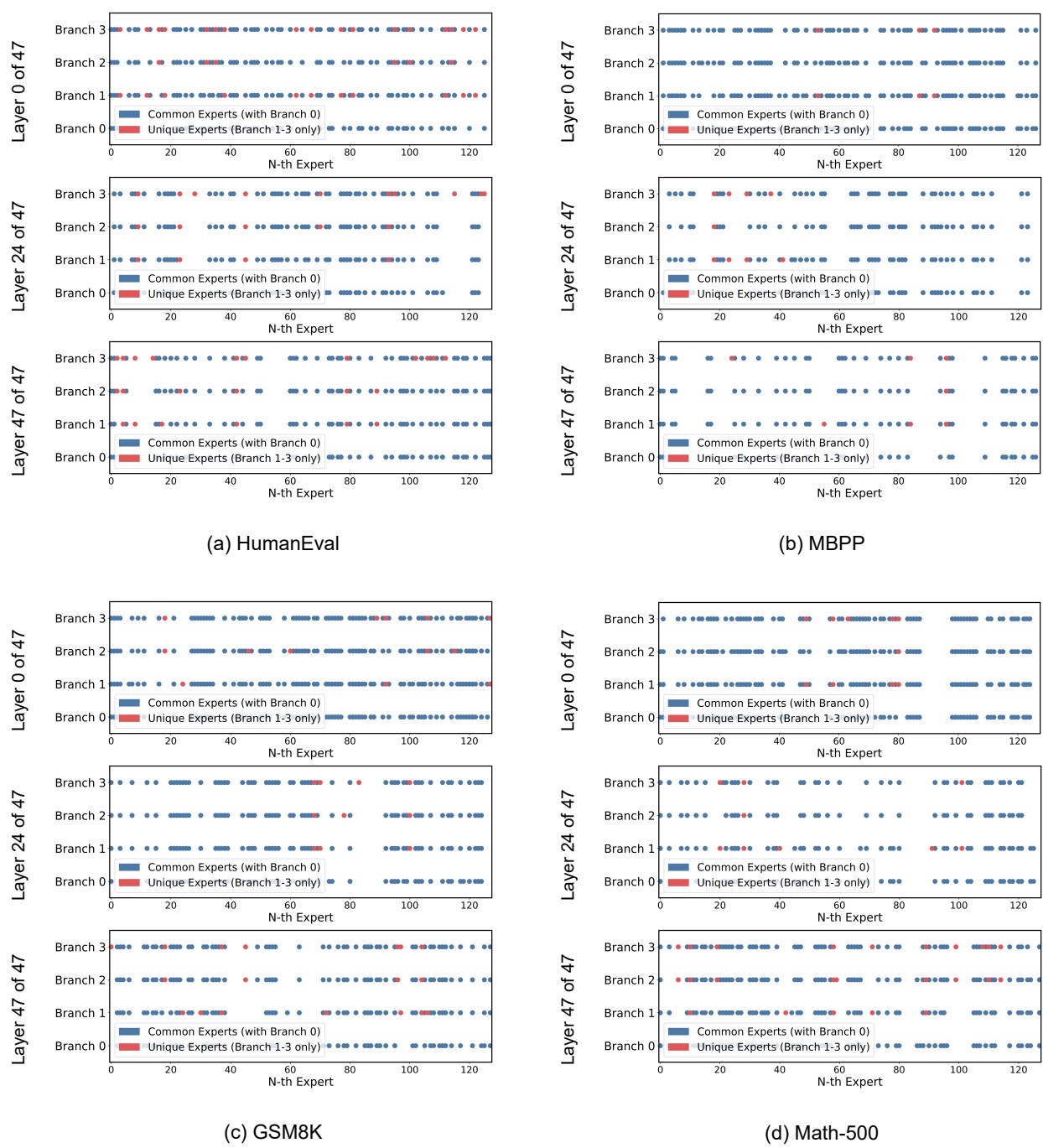

(a) HumanEval

(b) MBPP

(c) GSM8K

(d) Math-500

*Figure 11.* Expert activation with speculative exploration in SDAR 30B-A3B for responses from diverse benchmarks, measured at layer 0, 24 and 47 (of 47).

Experiments across diverse benchmarks show that, due to the high similarity among speculative branches, these branches tend to activate largely overlapping sets of experts, as shown in Figure 11. As a result, even when the effective decoding length increases by several times, aggressive speculative exploration introduces only a small number of additional experts, making multi-branch exploration particularly effective for accelerating MoE-based diffusion language models.

## B. Implementation of Limited Activation for Cold Tokens

Here, we detail the computation pipeline of our Limited Activation strategy for cold tokens, which incorporates a specially designed double-round routing mechanism.

---

**Algorithm 1** Limited Activation for Cold Tokens in SparseMoeBlock

---

**Require:** Hidden states $\mathbf{H}$, Past hidden states $\mathbf{H}_{past}$, Decoded index $\mathbf{D}$, Expert limit index $\mathbf{L}$, Number of experts $N$, Top-k value $k$

**Ensure:** Final hidden states $\mathbf{H}_{out}$, Router logits $\mathbf{R}$, Selected experts $\mathbf{E}_{sel}$

0: $\mathbf{M}_{compute} \leftarrow \neg\mathbf{D}$ {Compute mask: tokens needing computation}

0: $\mathbf{H}_{out}[\neg\mathbf{M}_{compute}] \leftarrow \mathbf{H}_{past}[\neg\mathbf{M}_{compute}]$ {Copy decoded tokens from cache}

0: **if** $\exists\, i : \mathbf{M}_{compute}[i] = \text{True}$ **then**

0:     $\mathbf{H}_{comp} \leftarrow \mathbf{H}[\mathbf{M}_{compute}]$ {Extract tokens to compute}

0:     **First-round Routing:**

0:     $\mathbf{R}_{comp} \leftarrow \text{Gate}(\mathbf{H}_{comp})$ {Compute router logits}

0:     $\mathbf{W} \leftarrow \text{Softmax}(\mathbf{R}_{comp})$ {Routing weights}

0:     $\mathbf{W}_{topk}, \mathbf{E}_{sel} \leftarrow \text{TopK}(\mathbf{W}, k)$ {Initial top-k selection}

0:     $\mathbf{L}_{comp} \leftarrow \mathbf{L}[\mathbf{M}_{compute}]$ {Expert limit mask for compute tokens}

0:     **if** $\exists\, i : \mathbf{L}_{comp}[i] = \text{True}$ **then** {Has cold tokens}

0:         **Collect Necessary Experts from Hot Tokens:**

0:         $\mathbf{I}_{hot} \leftarrow \{i : \mathbf{L}_{comp}[i] = \text{False}\}$ {Indices of hot tokens}

0:         $\mathbf{E}_{nec} \leftarrow \emptyset$ {Initialize necessary experts set}

0:         **if** $|\mathbf{I}_{hot}| > 0$ **then**

0:             $\mathbf{E}_{nec} \leftarrow \bigcup_{i \in \mathbf{I}_{hot}} \mathbf{E}_{sel}[i]$ {Union of experts selected by hot tokens}

0:         **end if**

0:         **Second-round Routing for Cold Tokens:**

0:         $\mathbf{M}_{invalid} \leftarrow \{e : e \notin \mathbf{E}_{nec}, e \in [0, N)\}$ {Invalid expert mask}

0:         $\mathbf{W}[:, \mathbf{M}_{invalid}] \leftarrow 0$ {Zero out invalid experts}

0:         $\mathbf{W}_{topk}, \mathbf{E}_{sel} \leftarrow \text{TopK}(\mathbf{W}, k)$ {Re-select from necessary experts only}

0:     **end if**

0:     **if** norm_topk_prob **then**

0:         $\mathbf{W}_{topk} \leftarrow \mathbf{W}_{topk} / \sum_{j=1}^{k} \mathbf{W}_{topk}[:, j]$ {Normalize}

0:     **end if**

0:     **Expert Computation:**

0:     $\mathbf{H}^{out}_{comp} \leftarrow \mathbf{0}$

0:     **for** $e = 0$ **to** $N - 1$ **do**

0:         $\mathbf{I}_e \leftarrow \{(i, j) : \mathbf{E}_{sel}[i, j] = e\}$ {Tokens assigned to expert $e$}

0:         **if** $|\mathbf{I}_e| > 0$ **then**

0:             $\mathbf{H}^{out}_{comp}[\mathbf{I}_e] \mathrel{+}= \text{Expert}_e(\mathbf{H}_{comp}[\mathbf{I}_e]) \odot \mathbf{W}_{topk}[\mathbf{I}_e]$

0:         **end if**

0:     **end for**

0:     $\mathbf{H}_{out}[\mathbf{M}_{compute}] \leftarrow \mathbf{H}^{out}_{comp}$

0:     $\mathbf{R}[\mathbf{M}_{compute}] \leftarrow \mathbf{R}_{comp}$

0: **end if**

0: **return** $\mathbf{H}_{out}, \mathbf{R}, \mathbf{E}_{sel}$ =0

---

