# OpenReview forum: "TEAM: Temporal–Spatial Consistency Guided Expert Activation for MoE Diffusion Language Model Acceleration"
_ICML.cc/2026/Conference — ICML 2026 regular_

### Official Review · Reviewer_wwai · 2026-03-02

**Soundness:** 3
**Presentation:** 3
**Significance:** 3
**Originality:** 3
**Overall Recommendation:** 4
**Confidence:** 4

**Summary:**

This paper investigates the inference efficiency challenges when integrating Mixture-of-Experts (MoE) architectures with Diffusion Large Language Models (dLLMs). The authors identify a fundamental mismatch where a large number of experts are activated per denoising step, yet only a few tokens are ultimately accepted, leading to substantial overhead. Leveraging the observed temporal and spatial consistency in expert routing, the authors propose TEAM, a framework comprising three strategies: Delayed Caching for Decoded Tokens (DCD), Speculative Exploration for Hot Tokens (SEH), and Limited Activation for Cold Tokens (LAC).

**Compliance With Llm Reviewing Policy:**

Affirmed.

**Key Questions For Authors:**

In the LAC strategy, if a cold token truly requires an inactivated expert for correctness, does forced rerouting to the active set cause long-term drift in generation quality?

The DCD strategy performs well in Refresh-free mode. Does this imply that intra-block attention in dLLMs is extremely stable in later iterations? Does this hold true for even larger models?

**Strengths And Weaknesses:**

Strengths:
- This may be the first study to specifically analyze expert activation characteristics in MoE dLLMs, revealing the efficiency mismatch between high expert activation and low token acceptance rates.
- The three proposed strategies are logically sound.
- The method shows significant end-to-end speedups.

Weaknesses:
- Experiments are only focused on the SDAR model.
- The classification of hot tokens depends on thresholds . While Table 3 provides ablation studies, it remains unclear if these fixed thresholds are effective for highly divergent tasks like creative writing.
- Although SEH increases computational branches during speculative exploration, the authors argue it is nearly "free" in memory-bound scenarios. A more detailed comparative analysis of FLOPs versus bandwidth utilization would be beneficial.

---

> ### Author Rebuttal · Authors · 2026-03-30
>
> Dear Reviewer wwai,
>
> We sincerely thank you for the positive evaluation and thoughtful questions regarding robustness and system behavior. Below are our responses to your concerns:
>
> ---
>
> ### 1. LAC Quality Preservation
> We acknowledge that modifying expert activations for cold tokens may alter decoding behavior, but it does not lead to long-term drift in generation quality. **First**, the classification of cold tokens is dynamic rather than fixed. At each iteration, tokens are re-identified based on updated decoding states and confidence scores, enabling adaptive partitioning across steps. **Second**, hot tokens cover most necessary experts due to high routing similarity among masked tokens. For the few experts uniquely activated by cold tokens, minor routing changes have negligible impact. This is also consistent with prior findings in autoregressive MoE models [1,2].
>
> [1] Zhong et al. "AdapMoE: Adaptive Sensitivity-based Expert Gating and Management for Efficient MoE Inference." (arXiv:2408.10284)
>
> [2] Wang et al. "BuddyMoE: Exploiting Expert Redundancy to Accelerate Memory-Constrained Mixture-of-Experts Inference." (arXiv:2511.10054)
>
> ---
>
> ### 2. DCD Stability Insight
> For the decoded tokens within a block, their attention and hidden states converge rapidly after being accepted and remain stable in subsequent iterations. As shown below, we analyze the average cosine similarity of decoded token representations across consecutive diffusion steps, indicating **strong representation stability and supporting the effectiveness of refresh-free caching**. Due to experimental constraints, we could not verify this on larger models. However, we expect this to generalize, as it stems from semantic stability rather than the model scale.
> ||HumanEval|MBPP|GSM8K|MATH-500
> ---|---|---|---|---
> Similarity|0.97|0.97|0.97|0.96
>
> ---
>
> ### 3. Generalization Capability
> We did not include LLaDA 2.0 experiments because its HuggingFace release lacks an official evaluation pipeline. Using lm-eval package, we obtained results consistent with prior unofficial reproductions [3], but significantly below reported performance, suggesting evaluation or implementation gaps that prevent fair assessment of TEAM.
> Nevertheless, we acknowledge the importance of generality and appreciate the reviewer’s suggestion. We performed **a preliminary integration of TEAM with LLaDA 2.0 using the SGLang engine** (results below). Full compatibility between TEAM and the engine requires more optimized operator-level implementations to minimize additional control overhead. We believe that due to the directness of our strategy, this can be achieved in the future. Despite these limitations, the results still show **consistent TPS improvements while maintaining accuracy**, supporting the general applicability of TEAM.
> |||HumanEval|MBPP|Average
> ---|---|---|---|---
> Vanilla|Score|79.88|81.26|80.57
> TEAM|Score|81.11|81.50|81.31
> Vanilla|TPS|387|258|322
> TEAM|TPS|440|296|368
>
> [3]https://github.com/preordinary/LLaDA2
>
> ---
>
> ### 4. Robust Token Partition
> We refine the sensitivity analysis (Tab.3 of our paper) by isolating the effects of two thresholds (https://anonymous.4open.science/r/icml26_rebuttal/table.png). TEAM maintains high accuracy across a wide range of configurations. Notably, **performance only degrades under extreme settings**, when the confidence threshold approaches 0.8 (close to the acceptance threshold of 0.95) or when the positional constraint becomes overly restrictive (distance < 3 compared to a block size of 32). This indicates that our token partition is robust across a broad parameter space, with parameters controlling efficiency rather than performance.
>
> ---
>
> ### 5. SEH Cost Analysis
> To clarify, our SEH strategy introduces minimal overhead per forward pass. Table below illustrates the operator-level computation and memory access w/ and w/o SEH, using a 1K KV cache and 64 activated experts, which are typical across benchmarks. Considering that typical GPU platforms such as the A100 have a compute-to-memory bandwidth ratio of 156 FLOPS/Byte, and more advanced GPUs like the H100 exceed 500 FLOPS/Byte, **the workload remains largely memory-bound**, preventing any increase in compute intensity from affecting inference latency. These findings are corroborated by our experimental results.
> w/o SEH|Compute(GFLOPs)|Memory(MB)|Ratio
> ---|---|---|---
> Q Proj|0.537|17.17|31.27
> K/V Proj|0.134|4.52|29.68
> Attention|1.096|19.86|55.19
> MoE FFN|2.434|611.86|3.98
> Others|2.433|610.29|3.99
>
> w/ SEH|Compute(GFLOPs)|Memory(MB)|Ratio
> ---|---|---|---
> Q Proj|2.147|18.35|117.03
> K/V Proj|0.536|5.51|97.52
> Attention|4.587|22.81|201.13
> MoE FFN|9.736|632|15.40
> Others|9.733|628|15.51
>
> ---
>
> We appreciate your constructive feedback and will incorporate these clarifications to improve the manuscript. We hope the responses above can address your concerns and contribute to a reconsideration of review score. Looking forward to discussing more with you.
>
> Best,
>
> Authors

---

> > ### Author Rebuttal · Reviewer_wwai · 2026-04-02
> >
> > NA

---

> > > ### Author Response · Authors · 2026-04-02
> > >
> > > Dear Reviewer wwai,
> > >
> > > Thank you for taking the time to review our rebuttal. We are glad that our responses have addressed your concerns. We appreciate your thoughtful evaluation and the helpful feedback that has improved our work.
> > >
> > > We hope these clarifications will be helpful in your final assessment.
> > >
> > > Best,
> > >
> > > Authors

---

### Official Review · Reviewer_7V9X · 2026-03-05

**Soundness:** 3
**Presentation:** 3
**Significance:** 3
**Originality:** 3
**Overall Recommendation:** 4
**Confidence:** 4

**Summary:**

This manuscript addresses the inference inefficiencies endemic to Diffusion Language Models (dLLMs) when scaled via Mixture-of-Experts (MoE) architectures. While MoE fundamentally decouples parameter count from active computational cost, applying it to the iterative, parallel decoding nature of diffusion models routinely triggers an "expert activation explosion." In a standard MoE dLLM, the continuous Markovian denoising process forces the gating network to re-evaluate and fetch experts for every token across every diffusion step, leading to catastrophic I/O bottlenecks and memory bandwidth saturation.

The authors identify a specific phenomenological behavior during the diffusion process—termed "temporal-spatial consistency"—whereby the expert routing decisions for individual tokens stabilize early in the denoising trajectory. Leveraging this observation, the paper proposes the TEAM framework, integrating intermediate token caching and speculative exploration to aggressively bypass redundant expert activations. The reported 2.2x acceleration in decoding latency without measurable perplexity degradation is technically sound.

However, while the localized algorithmic co-design is elegant, the manuscript suffers from severe isolation regarding its experimental baselines. By entirely circumventing comparisons with the industry-standard Autoregressive (AR) MoE paradigm, the broader utility of this acceleration framework remains ambiguous.

**Compliance With Llm Reviewing Policy:**

Affirmed.

**Final Justification:**

I have no more questions and I'd like to maintain my score.

**Key Questions For Authors:**

1.The authors must provide a direct, hardware-controlled comparison of generation latency (Time-To-First-Token and sustained Tokens-Per-Second) between the TEAM-accelerated MoE dLLM and an equivalent parameter-class Autoregressive MoE model (e.g., Mixtral 8x7B). This benchmark should be conducted on the same GPU hardware. Without this, the practical utility of the 2.2x relative speedup cannot be assessed.

2.Provide a detailed memory profile demonstrating how TEAM's caching mechanism impacts peak VRAM consumption as a function of sequence length (e.g., from 1K to 8K tokens). Plot the memory footprint against the vanilla MoE dLLM baseline.

3.Conduct an ablation study plotting the speculative acceptance rate against varying decoding temperatures and task complexities. Explicitly define the system's break-even point—at what acceptance rate does the TEAM framework result in a net latency regression due to recovery overhead?

4

**Limitations:**

yes

**Strengths And Weaknesses:**

Strengths

The primary scientific contribution of this manuscript is not the caching mechanism itself, but the empirical isolation and statistical validation of "temporal-spatial consistency" within the MoE gating networks of diffusion models.

Unlike autoregressive decoding, where token representations are generated sequentially and routing is strictly a function of the preceding context, diffusion decoding updates all tokens in parallel. The authors meticulously demonstrate that as the signal-to-noise ratio (SNR) increases during the reverse diffusion steps, the semantic representation of a given token crystallizes. Consequently, the probability distribution output by the MoE router sharpens rapidly. The observation that a token will overwhelmingly select the identical top-$k$ experts in step $T-1$ as it did in step $T$ provides a highly actionable foundation for architectural optimization.

Standard speculative decoding frameworks are generally confined to AR models, relying on draft models to predict sequential trajectories. TEAM adapts this philosophy to the continuous state space of diffusion. By caching the stabilized token representations and employing a speculative exploration mechanism, the framework allows the model to "skip" the computationally expensive expert routing and feed-forward execution for tokens that have reached representational consensus. This effectively bridges the parallel decoding advantages inherent to diffusion with the conditional computation advantages of MoE, a synthesis that is structurally elegant and computationally pragmatic.


 Weaknesses

The most critical flaw in this paper is its narrow comparative scope. The authors solely benchmark the TEAM-accelerated MoE dLLM against a vanilla MoE dLLM

While research into diffusion-based language modeling holds academic value (particularly for non-left-to-right generation tasks), the absolute dominant paradigm for LLM deployment is the Autoregressive (AR) model (e.g., LLaMA, Mixtral). If the authors claim to present a "Language Model Acceleration" framework, they must prove that accelerating a dLLM yields competitive absolute wall-clock metrics. Currently, an unoptimized dLLM is notoriously slow due to the requisite $N$ denoising steps. Achieving a 2.2x speedup on an inherently slow architecture does not automatically render it viable. There is no empirical evidence presented to show whether a TEAM-accelerated 8x7B dLLM can achieve higher tokens-per-second (TPS) than an equivalent 8x7B AR model served via highly optimized frameworks like vLLM or TensorRT-LLM.

The TEAM framework relies heavily on caching previously decoded tokens across the spatial-temporal grid to enable speculative exploration. In AR models, the KV cache grows linearly with sequence length, which is already the primary bottleneck in LLM serving.

In a diffusion context, caching intermediate representations across multiple denoising steps for parallel sequences introduces a distinct and potentially severe memory footprint. The manuscript completely omits a micro-level profiling of VRAM consumption. Does the caching mechanism introduce a super-linear memory scaling factor as sequence length increases? If the memory footprint expands significantly to accommodate the caching, the system will hit the memory capacity wall (Out-Of-Memory) far earlier, severely limiting the maximum batch size and, consequently, the peak aggregate throughput of the serving node.

Speculative execution frameworks are highly sensitive to the acceptance rate of the drafted/cached tokens. The paper evaluates TEAM on standard, likely deterministic benchmarks.

However, in high-temperature sampling regimes or during highly complex reasoning tasks (where token representations remain volatile until the final denoising steps), the temporal-spatial consistency is likely to degrade. If the acceptance rate drops below a certain threshold, the computational overhead of misprediction recovery (i.e., recalculating the expert routing and executing the correct experts) will exceed the latency saved by caching. The manuscript lacks a rigorous ablation study identifying this break-even point.

---

> ### Author Rebuttal · Authors · 2026-03-30
>
> Dear Reviewer 7V9X,
>
> We sincerely thank you for the insightful and system-level perspective, and for recognizing the technical soundness of our approach. We clarify the scope and practical considerations below:
>
> ---
>
> ### 1. Comparison with Autoregressive Models
> Traditional dLLMs are **compute-bound** due to full-sequence computation per iteration, limiting real speedup. In contrast, block diffusion changes the execution pattern by processing one block per step and caching completed blocks, keeping execution **memory-bound** and enabling trajectory reduction to translate into efficiency gains. Recent MoE dLLMs such as SDAR follow this paradigm and have reported efficiency surpassing the same-parameter autoregressive (AR) models under Pytorch framework.
> **To further assess practical efficiency in industrial scenarios**, we compare TTFT and TPS between Qwen3 30B-A3B and our SDAR 30B-A3B (trained from Qwen3) under the LMDeploy inference engine (results below). It demonstrates that even in a highly optimized framework, **our approach achieves comparable or better TPS than the AR model with the same architecture and parameter quantity**. We further clarify: (1) the relatively poor TTFT is caused by the fact that the prefill phase of SDAR does not generate the first token, but at the first decoding step within the block; (2) both vanilla SDAR and TEAM are still in early-stage integration with inference engines originally designed for AR models, and we expect further gains with improved pipeline design and operator-level optimizations.
> BS=1|Benchmark|HumanEval|MBPP|GSM8K|MATH-500
> ---|---|---|---|---|---
> Qwen|TTFT(s)|0.16|0.15|0.13|0.15
> Ours|TTFT(s)|0.25|0.32|0.27|0.27
> Qwen|TPS|113|120|127|133
> Ours|TPS|206|187|168|193
>
> BS=128|Benchmark|HumanEval|MBPP|GSM8K|MATH-500
> ---|---|---|---|---|---
> Qwen|TTFT(s)|0.75|0.64|0.42|0.43
> Ours|TTFT(s)|1.59|2.34|1.61|1.56
> Qwen|TPS|490|1114|928|1280
> Ours|TPS|1131|803|1352|1570
>
> ---
>
> ### 2. Bounded Memory Overhead
> As shown below, we provide a breakdown of memory usage. The additional memory introduced by DCD **does not grow with sequence length and remains strictly bounded**. This is due to the block diffusion paradigm widely adopted in modern dLLMs: bidirectional attention and parallel decoding are confined within each block, and once a block is decoded, its tokens and KV states are finalized and cached without further updates. Accordingly, **DCD only caches representations for decoded tokens within the current block**. The total cache size is therefore bounded by the block size (e.g., 32 tokens), resulting in negligible memory overhead.
> BS=1|Model|DCD-Cache|KV-Cache(1K)|KV-Cache(2K)|KV-Cache(4K)|KV-Cache(8K)
> ---|---|---|---|---|---|---
> Memory(G)|56.4|0.003|0.09|0.19|0.38|0.75
>
> ---
>
> ### 3. Speculative Efficiency Across Tasks & Temperatures
> As shown below, we provide additional results on HumanEval and a more challenging benchmark AIME, evaluating SEH under different temperature settings in terms of acceptance rate (Tokens-Per-Forward, TPF) and additional expert cost (Activations-Per-Forward, APF). **Across varying task complexities and temperatures**, SEH consistently maintains a high acceptance rate while introducing only minimal additional expert activations. This behavior is consistent with our analysis in the paper that masked tokens exhibit highly similar expert activation patterns, making it unlikely for SEH to reach a break-even point where speculative exploration becomes detrimental.
> Temperature|Configuration|APF(HumanEval)|TPF(HumanEval)|APF(AIME)|TPF(AIME)
> ---|---|---|---|---|---
> 0|w/o spec|53.2|2.36|54.7|2.84
> 0|w/ spec|59.2|3.49|60.4|4.21
> 0.5|w/o spec|54.3|3.92|54.3|4.70
> 0.5|w/ spec|59.2|4.48|58.1|5.84
> 0.8|w/o spec|53.4|2.96|55.2|3.31
> 0.8|w/ spec|59.3|3.90|60.5|4.58
>
> ---
>
> Thank you again for raising these important system-level considerations, which will help improve the clarity of our scope and evaluation. We hope the responses above can address your concerns and contribute to a reconsideration of review score. Looking forward to discussing more with you.
>
> Best,
>
> Authors

---

> > ### Author Rebuttal · Reviewer_7V9X · 2026-04-03
> >
> > Thank you for your rebuttal. I have no more questions and I'd like to maintain my score.

---

> > > ### Author Response · Authors · 2026-04-03
> > >
> > > Dear Reviewer 7V9X,
> > >
> > > Thank you for taking the time to review our rebuttal. We are glad that our responses have addressed your concerns. We appreciate your thoughtful evaluation and the helpful feedback that has improved our work.
> > >
> > > We hope these clarifications will be helpful in your final assessment.
> > >
> > > Best,
> > >
> > > Authors

---

### Official Review · Reviewer_ETMi · 2026-03-11

**Soundness:** 2
**Presentation:** 3
**Significance:** 3
**Originality:** 3
**Overall Recommendation:** 5
**Confidence:** 5

**Summary:**

The paper focuses on the low inference efficiency that arises when combining the MoE architecture with diffusion large language models (dLLMs). It first analyzes the key challenges of this integration, identifying that the decoding process in dLLMs introduces substantial redundant computation and resource overhead. To address this issue, the paper proposes an acceleration framework called TEAM. Leveraging the temporal–spatial consistency in the block-wise decoding of dLLMs, TEAM applies a delayed caching mechanism for already-decoded tokens to eliminate redundant expert activations. For “hot” tokens with a high probability of acceptance, it performs multi-branch speculative exploration to increase the token acceptance rate per iteration. For “cold” tokens with a low acceptance probability, it restricts the activation range of experts through a two-round expert routing mechanism. Experiments on the SDAR 30B-A3B demonstrate the effectiveness of the TEAM framework, achieving up to 2.2× and an average 1.94× inference speedup, while maintaining nearly unchanged task performance.

**Compliance With Llm Reviewing Policy:**

Affirmed.

**Final Justification:**

- The authors have addressed the majority of the raised To further strengthen the manuscript, the authors may consider adding more detailed descriptions and analyses for the new results and contents in the revision. Given this, I will adjust my rating.

**Key Questions For Authors:**

1. The paper mentions that LLaDA 2.0 was not evaluated because the HuggingFace evaluation pipeline is not available. Could the authors clarify the specific reason for this limitation? For example, is it due to missing model checkpoints, incompatibility with the evaluation scripts, or other implementation constraints?
2. Please provide a more detailed response to the issues raised in the experimental analyses mentioned in the weaknesses section. In particular, it would be helpful to clarify the design choices in SEH and the explanation for the performance difference between refresh-8 and refresh-free in DCD.
3. The observation that a specific group of experts dominates the decoding process for most masked tokens, while other experts contribute only marginally or are rarely activated, is very interesting. What could be the underlying reason for this phenomenon?

**Limitations:**

The paper could clarify that TEAM is currently optimized for single-GPU inference scenarios. It may not directly apply to settings involving multi-GPU expert parallelism or distributed inference, where expert routing and communication overhead follow different patterns. Explicitly stating this limitation would help better define the scope and applicability of the proposed method.

**Strengths And Weaknesses:**

Strengths
1. notable originality.
To the best of my knowledge, this is the first work that systematically studies the expert activation problem arising from the combination of MoE and diffusion large language models (dLLMs). The proposed TEAM framework provides a dedicated acceleration solution tailored to this paradigm, filling an important gap in the current literature.
2. Clear and well-structured analysis.
The paper presents a clear and logical analysis. Preliminary experiments effectively reveal the inference inefficiency in the MoE+dLLM setting and identify the key insight of temporal–spatial consistency in expert activation. This observation provides a solid basis for the subsequent method design.
3. Reasonable method design.
The framework introduces three complementary strategies—DCD, SEH, and LAC—to handle different types of tokens within a decoding block. The overall design is well aligned with the characteristics of the MoE+dLLM architecture, and the technical pipeline appears coherent without obvious logical issues.
4. Strong practical value.
TEAM is a plug-and-play framework that does not require model retraining and can be applied directly to the MoE+dLLM inference pipeline. Experiments on SDAR 30B-A3B demonstrate up to 2.2× speedup on a single GPU with negligible performance degradation, indicating strong practical applicability.

Weaknesses
1. Limited validation of framework generalization.
The experiments are conducted only on SDAR 30B-A3B. The framework is not evaluated on other MoE+dLLM models[1,2], such as LLaDA 2.0 or LLaDA-MoE. Therefore, the generality and adaptability of the proposed method remain unclear.
2. Some experimental analyses lack depth.
In SEH, the paper does not evaluate combinations of non-aligned candidate tokens for hot tokens, making it difficult to determine whether the current configuration is optimal. In DCD, the observation that refresh-8 performs worse than refresh-free is only described empirically, with limited analysis explaining the underlying reason.
3. Issues in hyperparameter sensitivity analysis.
In the sensitivity study, the confidence threshold for identifying hot tokens and the spatial distance parameter are varied simultaneously. This design makes it difficult to isolate and understand the independent effect of each parameter. A more controlled analysis would improve the interpretability of the results.
Reference
[1] Zhu, Fengqi, et al. "Llada-moe: A sparse moe diffusion language model." arXiv preprint arXiv:2509.24389 (2025).
[2] Bie, Tiwei, et al. "Llada2. 0: Scaling up diffusion language models to 100b." arXiv preprint arXiv:2512.15745 (2025).

---

> ### Author Rebuttal · Authors · 2026-03-30
>
> Dear Reviewer ETMi,
>
> We sincerely thank you for the highly positive evaluation, especially for recognizing the originality and practical value of our work. We address the concerns on generalization and analysis below:
>
> ---
>
> ### 1. Generalization Across Models
> LLaDA-MoE employs global bidirectional attention, which lags behind the block diffusion paradigm in both accuracy and efficiency. LLaDA 2.0 adopts block-wise decoding, but its HuggingFace release lacks an official evaluation pipeline. Using lm-eval package, we obtained results consistent with prior unofficial reproductions [1], but significantly below reported performance, suggesting evaluation or implementation gaps that prevent fair assessment of TEAM.
>
> Nevertheless, we acknowledge the importance of generality and appreciate the suggestion. We performed **a preliminary integration of TEAM with LLaDA 2.0 using the SGLang engine** (results below). We identify two factors limiting the observed speedup: (1) the highly optimized inference engine reduces kernel launch overhead, weakening the benefit of fewer expert activations; (2) full compatibility between TEAM and the engine requires more optimized operator-level implementations to minimize additional control overhead (e.g., altering the original CUDA execution graph in SEH). We believe that due to the directness of our strategy, this can be achieved in the future. Despite these limitations, we still observe **consistent TPS improvements while maintaining accuracy**, supporting the general applicability of TEAM.
> |||HumanEval|MBPP|Average
> ---|---|---|---|---
> Vanilla|Score|79.88|81.26|80.57
> TEAM|Score|81.11|81.50|81.31
> Vanilla|TPS|387|258|322
> TEAM|TPS|440|296|368
>
> [1] https://github.com/preordinary/LLaDA2
>
> ---
>
> ### 2. Further Experimental Analysis
> **SEH Design Choice.** As shown below, we analyze the acceptance probability of the top-3-confidence candidate tokens in the next iteration. The 2nd candidate has a relatively low probability of being directly accepted, while the 3rd one is rarely accepted. This indicates that constructing more diverse branches is inefficient. In contrast, aligned token combinations enable chained verification, thereby improving efficiency.
> ||HumanEval|MBPP|GSM8K|MATH-500
> |---|---|---|---|---
> Top-1|0.71|0.68|0.70|0.71
> Top-2|0.32|0.29|0.36|0.39
> Top-3|0.20|0.17|0.25|0.27
>
> **DCD Refresh Behavior.** Minor performance differences across refresh settings are expected due to variations in decoding trajectories. In fact, the stronger performance of Refresh-free over Refresh-8 just indicates that DCD influences the decoding trajectory, but does not inherently degrade model performance. To further support this, we analyze the average cosine similarity of decoded token representations across consecutive diffusion steps, indicating strong representation stability and supporting the effectiveness of refresh-free caching.
> ||HumanEval|MBPP|GSM8K|MATH-500
> ---|---|---|---|---
> Similarity|0.97|0.97|0.97|0.96
>
> ---
>
> ### 3. Expert Dominance Explanation
> We thank the reviewer for highlighting this key point. **The dominance arises from the high similarity among model inputs for masked tokens.** All masked tokens share the same [M] symbol and are mapped to identical token embeddings. The only source of variation comes from positional encodings, which are relatively minor within a block. As shown below, we compute the average pairwise cosine similarity of hidden states among all masked tokens within a block at each decoding layer, and then report the highest and the average similarity in all layers. Such a high degree of similarity indicates that this homogeneity persists throughout the entire decoding process, leading to similar routing decisions.
> ||HumanEval|MBPP|GSM8K|MATH-500
> ---|---|---|---|---
> Highest|0.99|0.98|0.99|0.99
> Average|0.86|0.84|0.86|0.86
>
> ---
>
> ### 4. Isolated Sensitivity Analysis
> We agree that varying parameters jointly limits interpretability. We have conducted additional controlled experiments isolating each parameter to improve clarity  (https://anonymous.4open.science/r/icml26_rebuttal/table.png). The results demonstrate that **TEAM maintains consistently high accuracy across a wide range of configurations**. Notably, performance only degrades under extreme settings, when the confidence threshold approaches 0.8 (close to the acceptance threshold of 0.95) or when the positional constraint becomes overly restrictive (distance < 3 compared to a block size of 32).
>
> ---
>
> ### 5. Scope and Limitations
> Thanks for your suggestion. We will clarify that TEAM is currently optimized for single-GPU inference, and that extending it to distributed settings requires further system-level design.
>
> ---
>
> We appreciate your detailed and constructive feedback, and believe the suggested improvements will further strengthen the paper. We hope the responses above can address your concerns and contribute to a reconsideration of review score. Looking forward to discussing more with you.
>
> Best,
>
> Authors

---

> > ### Author Rebuttal · Reviewer_ETMi · 2026-04-03
> >
> > - The authors have addressed the majority of the raised To further strengthen the manuscript, the authors may consider adding more detailed descriptions and analyses for the new results and contents in the revision. Given this, I will adjust my rating.

---

> > > ### Author Response · Authors · 2026-04-03
> > >
> > > Dear Reviewer ETMi,
> > >
> > > Thank you for your careful evaluation and for updating your rating. We are glad that our responses have addressed your concerns. We also appreciate your suggestion on further strengthening the presentation, and we will incorporate more detailed descriptions and analyses of the new results in the revision.
> > >
> > > Best,
> > >
> > > Authors

---

### Official Review · Reviewer_YBYA · 2026-03-13

**Soundness:** 2
**Presentation:** 3
**Significance:** 2
**Originality:** 2
**Overall Recommendation:** 4
**Confidence:** 3

**Summary:**

This paper proposes TEAM, a plug-and-play acceleration framework for Mixture-of-Experts (MoE) diffusion large language models (dLLMs). Overall, a general context investigated by the paper is improving the inference efficiency of diffusion-based language models that adopt MoE architectures under block-wise diffusion decoding. Overall, a major problem examined by the study is that naive integration of MoE with diffusion decoding leads to excessive expert activations per denoising iteration, since many tokens independently route to different experts while only a small subset of tokens are ultimately accepted, resulting in high memory access and communication overhead. To address this inefficiency, the authors analyze expert activation behavior in MoE dLLMs and identify strong temporal consistency across diffusion iterations and spatial consistency across token positions. Based on these observations, they propose TEAM, which introduces three coordinated expert activation strategies within each decoding block: (1) Delayed Caching for Decoded tokens (DCD), which caches previously accepted tokens and avoids repeatedly activating experts for tokens whose representations have stabilized; (2) Speculative Exploration for Hot tokens (SEH), which identifies masked tokens likely to be accepted using confidence and spatial proximity heuristics and performs multi-branch speculative decoding to increase token acceptance per iteration; and (3) Limited Activation for Cold tokens (LAC), which constrains expert routing for unlikely tokens by restricting their expert selection to a necessary expert set determined by decoded and hot tokens through a two-round routing procedure. These mechanisms collectively reduce redundant expert activation while increasing token acceptance per forward pass, thereby improving the decoding efficiency of MoE diffusion language models without modifying model training.

**Compliance With Llm Reviewing Policy:**

Affirmed.

**Final Justification:**

My concerns have been addressed. I would like to keep my score.

**Key Questions For Authors:**

A particularly interesting aspect of the method is that it effectively partitions tokens into different groups, such as, already decoded tokens, hot masked tokens that are likely to be accepted soon, and cold masked tokens that are unlikely to be accepted in the near future. Given this token taxonomy, I wonder whether the efficiency analysis could be refined beyond aggregate APF / TPF / ATP metrics. For example, could the authors decompose expert activation and latency savings by token category, such as: expert activations attributable to decoded tokens before vs. after DCD, marginal APF increase introduced by hot-token speculative branches together with their TPF gain, and expert activations avoided for cold tokens due to restricted routing in LAC? Such a breakdown would make it much easier to understand which component contributes most to the final speedup, and would also better validate the paper’s central claim that different token categories require different expert activation policies.

**Limitations:**

The paper does not discuss its limitations with sufficient clarity.

**Strengths And Weaknesses:**

Strengths

The paper identifies a practically meaningful empirical phenomenon in MoE diffusion decoding: expert usage is not arbitrary across denoising steps, but exhibits noticeable stability after token acceptance and substantial regularity across neighboring masked positions. This observation is valuable beyond the specific TEAM design, because it provides a useful inductive bias for constructing inference-time heuristics in MoE dLLMs.


Weaknesses

The method appears somewhat sensitive to hyperparameter choices. In particular, the partition of masked tokens into hot and cold categories depends on manually chosen thresholds such as the confidence threshold and distance constraint, and the paper’s own sensitivity study suggests that overly aggressive narrowing of the hot-token set can noticeably hurt model quality. This makes the method look less robust than the main results may initially suggest.

---

> ### Author Rebuttal · Authors · 2026-03-30
>
> Dear Reviewer YBYA,
>
> We sincerely thank you for highlighting the practical importance of temporal-spatial consistency. We also appreciate the insightful suggestions regarding robustness and finer-grained analysis. Below are our responses to your concerns:
>
> ---
>
> ### 1. Robustness of Hyperparameter Settings
>
> We clarify that the hyperparameters used to distinguish hot/cold tokens are not arbitrarily tuned, but **control a trade-off between accuracy and sparsity**, grounded in the observed temporal-spatial consistency. We refine the sensitivity analysis (Tab.3 of our paper) by isolating the effects of two thresholds (https://anonymous.4open.science/r/icml26_rebuttal/table.png). It shows that TEAM maintains consistently high accuracy across a wide range of configurations. Notably, **performance only degrades under extreme settings**, when the confidence threshold approaches 0.8 (close to the acceptance threshold of 0.95) or when the positional constraint becomes overly restrictive (distance < 3 compared to a block size of 32). It indicates that TEAM is robust across a broad parameter space, and our settings primarily govern the optimal efficiency rather than introducing performance loss.
>
> ---
>
> ### 2. Refined Analysis by Token Category
>
> We agree that a finer-grained breakdown is valuable. We have conducted additional analysis across token categories.
>
> (1) **DCD: Reduced Activations for Decoded Tokens.** We analyze the average number of activated experts per layer when applying DCD, along with the subset attributable to decoded tokens. DCD reduces total expert activations by approximately 40%, and reduces nearly 60% experts activated only by decoded tokens, confirming effective removal of redundant activations.
> ||w/o DCD||w/ DCD||
> ---|---|---|---|---
> Benchmark|Sum of Act. experts|Act. only by decoded tokens|Sum of Act. experts|Act. only by decoded tokens
> HumanEval|53.34|25.34|31.57|10.43
> MBPP|49.59|21.96|27.84|8.76
> GSM8K|59.11|29.90|36.39|12.69
> MATH-500|57.90|31.16|36.33|14.35
> Average|54.99|27.09|33.03|11.56
>
> (2) **SEH: Efficiency of Speculative Exploration.** We examine the additional activations introduced by SEH and the corresponding acceptance gains. Due to the similarity in branches and activation patterns (Fig.4 of our paper), SEH incurs only a 12% increase in APF, while achieving a 1.6× improvement in TPF. This leads to 30% fewer experts activated per decoded token on average.
> ||w/o SEH|||w/ SEH|||
> ---|---|---|---|---|---|---
> Benchmark|APF|TPF|APT|APF|TPF|APT
> HumanEval|53.34|2.91|18.33|59.56|4.87|12.23
> MBPP|49.59|2.74|18.10|56.72|4.52|12.55
> GSM8K|59.11|3.16|18.71|66.03|5.00|13.21
> MATH-500|57.90|3.74|15.48|64.94|5.72|11.35
> Average|54.99|3.14|17.66|61.81|5.03|12.34
>
> (3) **LAC: Constrained Activation for Cold Tokens.** We further analyze the effect of LAC on expert activations for masked tokens. To avoid overestimating necessary activations from decoded tokens, we build this analysis on top of DCD. LAC removes experts activated solely by cold tokens, further shrinking the activation set.
> ||w/o LAC||w/ LAC||
> ---|---|---|---|---
> Benchmark|Act. only by masked tokens|Act. only by cold tokens|Act. only by masked tokens|Act. only by cold tokens
> HumanEval|13.82|4.23|9.03|0
> MBPP|12.60|3.31|7.41|0
> GSM8K|15.11|5.26|9.08|0
> MATH-500|13.68|4.65|9.26|0
> Average|13.80|4.36|8.70|0
>
> ---
>
> ### 3. Expanded Limitations
>
> We agree that the limitations can be more clearly stated in the revision. Our current design targets scenarios that are highly sensitive to decoding speed and tail latency, as well as edge platforms with constrained hardware resources, where single-GPU deployment is a practical and relevant setting. More complex routing and communication patterns may exist in distributed scenarios, which we leave for future work.
>
> ---
>
> Thank you again for the constructive feedback, which will help improve both the clarity and completeness of the paper. We hope the responses above can address your concerns and contribute to a reconsideration of review score. Looking forward to discussing more with you.
>
> Best,
>
> Authors

---

### Decision · Program_Chairs · 2026-04-30

**Decision:**

Accept (regular)

**Comment:**

## Summary
This paper proposes TEAM, a plug-and-play acceleration framework for Mixture-of-Experts (MoE) diffusion large language models (dLLMs). The framework exploits temporal and spatial consistency in expert routing to reduce redundant expert activations through three coordinated strategies: Delayed Caching for Decoded tokens (DCD), Speculative Exploration for Hot tokens (SEH), and Limited Activation for Cold tokens (LAC). Experiments on SDAR 30B-A3B report up to 2.2x inference speedup with minimal quality degradation. Reviewers acknowledged the practical relevance of the problem and the reasonable algorithmic design, but raised significant concerns about the scope of evaluation, missing baselines, and insufficient discussion of limitations.

## Justification for acceptance

The paper is very well written the proposed method is interesting and relevant to the ICML community. All the reviewers are unanimously positive about this paper and I would recommend this paper for acceptance.